# Prescription-based prediction of baseline mortality risk among older men

**Rolf Gedeborg**[1]*, **Hans Garmo**[2,3,4], **David Robinson**[5], **Pär Stattin**[2]

**1** Dept. of Surgical Sciences, Anesthesiology and Intensive Care, Uppsala University, Uppsala, Sweden, **2** Dept. of Surgical Sciences, Urology, Uppsala University, Uppsala, Sweden, **3** Translational Oncology and Urology Research (TOUR), School of Cancer and Pharmaceutical Sciences, King's College London, Guy's Hospital, London, United Kingdom, **4** Regional Cancer Center Uppsala Örebro, Uppsala University Hospital, Uppsala, Sweden, **5** Department of Urology, Ryhov Hospital, Jönköping, Sweden

* rolf.gedeborg@surgsci.uu.se

## Abstract

### Background

Understanding the association between patients' history of prescribed medications and mortality rate could optimize characterization of baseline risk when the Charlson Comorbidity Index is insufficient.

### Methods

Using a Swedish cohort of men selected randomly as controls to men with prostate cancer diagnosed 2007–2013, we estimated the association between medications prescribed during the previous year and mortality rates, using Cox regression stratified for age.

### Results

Among the 326,450 older men with median age of 69 years included in this study, 73% were categorized as free of comorbidity according to the Charlson Comorbidity Index; however, 84% had received at least one prescription during the year preceding the follow-up. This was associated with a 60% overall increase in mortality rate (hazard ratio [HR] = 1.60, 95% confidence interval [CI] 1.56 to 1.64). Some drugs that were unexpectedly associated with mortality included locally acting antacids (HR = 4.7, 95% CI 4.4 to 5.1), propulsives (HR = 4.7, 95% CI 4.4 to 5.0), vitamin A and D (HR = 4.6, 95% CI 4.3 to 4.9), and loop diuretics, for example furosemide (HR = 3.7; 95% CI 3.6 to 3.8). Thiazide diuretics, however, were only weakly associated with a mortality risk (HR = 1.5; 95% CI 1.4 to 1.5). Surprisingly, only weak associations with mortality were seen for major cardiovascular drug classes.

### Conclusions

A majority of older men had a history of prescribed medications and many drug classes were associated with mortality rate, including drug classes not directly indicated for a specific comorbidity represented in commonly used comorbidity measures. Prescription history can improve baseline risk assessment but some associations might be context-sensitive.

**Data Availability Statement:** Data used in the present study was extracted from the Prostate Cancer Database Sweden (PCBaSe), which is based on the National Prostate Cancer Register (NPCR) of Sweden. The data can be retrieved after

application made to any of the steering groups of NPCR and PCBaSe. For detailed information, please see www.npcr.se/in-english, where registration forms, manuals, and annual reports from NPCR are available alongside a full list of publications from PCBaSe. The code used for the present study analyses can be provided on request (contact via hans.garmo@kcl.ac.uk).

**Funding:** Pär Stattin received funding from the Swedish Cancer Society 17-0474 and the Swedish Research Council 17-00847. David Robinson received funding from the Clinical Cancer Research Foundation in Jönköping, Sweden. The funders had no role in study design, data collection and analysis, decision to publish, or preparation of the manuscript.

**Competing interests:** The authors have declared that no competing interests exist.

# Introduction

Adjusting for baseline comorbidity is a common practice in epidemiological studies. For example, using the Charlson Comorbidity Index (CCI) is a well-established strategy in registry-based studies [1, 2]. While the CCI might perform well in some situations, a significant number of patients score very low or zero on the CCI despite having a comorbidity, suggesting that the CCI might not be sufficiently discriminative. Data from a prescription registry could potentially provide additional information regarding patients' baseline health status and associated mortality risk. Prescriptions for medication provide indirect information on patients' current medical condition. Patients' prescription history tends to be readily available and not dependent on manual coding procedures. Concurrently, prescriptions issued as part of unspecific, symptomatic treatment or preventive approach might provide supplementary information that is independent of any specific condition.

The association between different aspects of prescription history and mortality has been comprehensively evaluated in several previous studies [3, 4]. Prescription history, as a measure of comorbidity, has also been described previously [5]. A common approach has been to take a number of relevant comorbidities as the starting point and identify the Anatomical Therapeutic Chemical (ATC) codes for drugs used to treat these conditions [6]. This may, however, inadvertently overlook important information regarding the patients' baseline mortality risk, suggesting a need for an approach involving non-specific prescriptions. This study aimed to broadly explore prescription history, which includes all available prescribed drugs, to describe its potential usefulness for characterization of baseline mortality risk.

Our hypothesis was that an ATC-code might provide relevant information regarding the patient's health status even if the indication is unknown or not associated with mortality risk per se. To investigate this hypothesis, we assessed associations between prescriptions for ATC-codes at a pharmaceutical subgroup level and mortality in a population of elderly men.

# Methods

## Study population

In the Prostate Cancer Database Sweden (PCBaSe) [7], National Prostate Cancer Register (NPCR) of Sweden (NPCR) has been linked to other registries, including Swedish Cancer Registry [8], Cause of Death Registry [9], Prescribed Drug Registry [10], and National Patient Registry (PAR) [11] through the unique Swedish Personal Identity Number (PIN) [12]. The PCBaSe 4.1 included men diagnosed with prostate cancer between 1998 and 2016 together with five men free of prostate cancer randomly selected from the general population matched for birth year and county of residence [13]. For this study, we used control men for prostate cancer cases diagnosed between 2007 and 2013. The start of follow-up for a control man was the date of diagnosis of prostate cancer for his matched case. The study was approved by the Research Ethics Board in Uppsala that waived the informed consent requirement.

## Covariates

The CCI was calculated based on discharge diagnoses from hospitalizations and specialist outpatient visits, extracted from PAR for the 10-year period preceding the start of follow-up [2].

## Filled prescriptions

The National Prescribed Drug Registry is compulsory, nationwide, and run by the Swedish Board of Health and Welfare [10]. It contains detailed and comprehensive information on all prescribed and dispensed drugs in Sweden, and includes the unique person identity number of

the patient since 1 July 2005. It does not have information on drugs administered in the hospital setting. Filled prescriptions were extracted during the 1-year period preceding the individual's follow-up. The prescribed drugs were categorized using the first level (anatomical main group) of the ATC classification system for descriptive purposes, and with the third level (pharmacological subgroup) for the main analyses. The pharmacological subgroups represented in the PCBaSe study population are listed in S1 Table.

## Statistics

The men were followed until the date of emigration, death, or end of follow-up (31 December 2017), whichever came first. The association with mortality was described using Kaplan–Meier survival curves. Results were presented for the 100 pharmacological subgroups with the highest number of deaths. Hazard ratios (HR) and corresponding 95% confidence intervals (CI) were calculated using univariable Cox proportional hazard models stratified by age (0–50, 51–60, 61–65, 66–70, /. . ./, 96–110 years). HRs were calculated for the respective ATC pharmacological subgroups using the subjects without any prescription as a reference. In the subgroup analysis, we restricted the population to that of men with CCI = 0. To avoid violating the assumption of proportional hazards, the analysis was stratified for age rather than being adjusted by it as a continuous variable. We, therefore, also performed a sensitivity analysis without stratification and instead adjusted for age as a continuous and quadratic term in the Cox model.

## Results

There were 326,450 men with a median age of 69 years (interquartile range 63–75) included in the PCBaSe as controls to cases diagnosed with prostate cancer between 2007 and 2013. Their characteristics in relation to their CCI are described in Table 1. While 73% were categorized as free of comorbidity according to the CCI, 84% had received at least one prescription during the year before the start of follow-up. These men tended to be older and had a lower educational status than that of the minority who did not receive any prescriptions during the year preceding the follow-up. Receiving at least one prescription was associated with a 60% increase in mortality rate (HR = 1.60, 95% CI 1.56 to 1.64) than not receiving any prescription.

The most common prescription was for drugs affecting the cardiovascular system (anatomical main group C; 58%), followed by drugs for the blood and blood-forming organs (group B; 41%), nervous system (group N; 38%), and alimentary system (group A; 37%). In men with CCI = 0, indicating no comorbidity, the most common prescriptions belonged to the anatomical main groups as follows: cardiovascular system (47%), nervous system (31%), alimentary system (27%), and blood and blood-forming organs (26%).

The associations between each pharmacological subgroup (four positions of the ATC code) and mortality are presented in Figs 1–5. The survival curves display the crude survival in the respective ATC pharmacological subgroup compared to all other men in the study population with at least one registered prescription. The HRs are age-adjusted.

### Alimentary tract and metabolism (ATC-group A)

Treatment with antacids was associated with long-term mortality rate (Fig 1). The association was stronger (HR = 4.7, 95% CI 4.4 to 5.1) for locally acting antacids (A02A) than for the more prevalent systemic antacids such as proton pump inhibitors and H2-receptor antagonists (HR = 2.06, 95% CI 2.00 to 2.12) (A02B). Use of propulsives (A03F), such as metoclopramide, indicated for nausea and vomiting, was also associated with long-term mortality (HR = 4.7,

**Table 1. Characteristics of prostate cancer-free men.**

| | CCI = 0 (n = 237,515) | CCI = 1 (n = 44,127) | CCI = 2 (n = 23,206) | CCI = 3 (n = 10,328) | CCI ≥ 4 (n = 11,274) | All (n = 326,450) |
|---|---|---|---|---|---|---|
| **Age, % (n)** | | | | | | |
| ≤65 | 42 (98779) | 22 (9911) | 17 (3902) | 12 (1242) | 13 (1450) | 35 (115284) |
| 66–75 | 40 (94574) | 42 (18358) | 39 (8962) | 36 (3750) | 37 (4136) | 40 (129780) |
| >75 | 19 (44162) | 36 (15858) | 45 (10342) | 52 (5336) | 50 (5688) | 25 (81386) |
| **Number of prescribed drugs [a], % (n)** | | | | | | |
| 0 | 22 (51206) | 2 (838) | 2 (451) | 1 (80) | 1 (102) | 16 (52677) |
| 1 | 18 (43936) | 3 (1404) | 3 (718) | 1 (99) | 1 (114) | 14 (46271) |
| 2 | 18 (43922) | 14 (6265) | 9 (1997) | 4 (459) | 3 (322) | 16 (52965) |
| 3 | 16 (37560) | 21 (9077) | 16 (3754) | 12 (1225) | 8 (920) | 16 (52536) |
| 4 | 12 (27746) | 21 (9467) | 21 (4900) | 20 (2035) | 16 (1766) | 14 (45914) |
| 5 | 7 (17445) | 17 (7514) | 20 (4570) | 23 (2400) | 22 (2472) | 11 (34401) |
| 6 | 4 (9307) | 12 (5138) | 15 (3485) | 18 (1891) | 21 (2369) | 7 (22190) |
| 7+ | 3 (6393) | 10 (4424) | 14 (3331) | 21 (2139) | 28 (3209) | 6 (19496) |
| **ATC-codes, % (n)** | | | | | | |
| A- | 27 (63572) | 52 (23019) | 67 (15438) | 78 (8066) | 86 (9701) | 37 (119796) |
| B- | 26 (62707) | 78 (34622) | 78 (18210) | 86 (8895) | 87 (9755) | 41 (134189) |
| C- | 47 (110676) | 86 (37952) | 85 (19713) | 91 (9381) | 91 (10292) | 58 (188014) |
| G- | 15 (36121) | 21 (9226) | 22 (5042) | 23 (2370) | 22 (2536) | 17 (55295) |
| H- | 8 (18160) | 15 (6445) | 18 (4112) | 21 (2214) | 27 (3042) | 10 (33973) |
| J- | 26 (61149) | 36 (15963) | 42 (9857) | 49 (5054) | 58 (6522) | 30 (98545) |
| L- | 1 (3019) | 3 (1257) | 4 (1021) | 6 (607) | 8 (860) | 2 (6764) |
| M- | 24 (56917) | 28 (12534) | 29 (6768) | 31 (3207) | 35 (3925) | 26 (83351) |
| N- | 31 (73126) | 51 (22487) | 60 (14032) | 70 (7267) | 78 (8741) | 38 (125653) |
| R- | 11 (25790) | 19 (8211) | 22 (5041) | 27 (2777) | 30 (3334) | 14 (45153) |
| S- | 15 (34747) | 20 (8825) | 22 (5163) | 25 (2539) | 26 (2947) | 17 (54221) |

The data is presented stratified by the level of comorbidity as indicated by the Charlson Comorbidity Index (CCI).

[a] One subject could have prescriptions from several anatomical main groups. The numbers and percentages therefore do not add up to the number of subjects and the sum of percentages exceeds 100%.

95% CI 4.4 to 5.0). The association was particularly strong for serotonin antagonists (A04A; HR = 10.2, 95% CI 9.4 to 11.1).

Other drugs that were associated with death rate included intestinal antiinfectives (A07A) indicated for treatment of Clostridium difficile diarrhea (HR = 3.8, 95% CI 3.6 to 4.1), and antipropulsives (A07D; HR = 3.0, 95% CI 2.8 to 3.1). Vitamins and minerals (A11–A12) showed an association with mortality, specifically, vitamin A and D (HR = 4.6, 95% CI 4.3 to 4.9), and potassium (HR = 3.2, 95% CI 3.1 to 3.4). Drugs used in diabetes (A10) were associated with mortality rate; insulins were associated with a risk increase (A10A; HR = 3.2, 95% CI 3.1 to 3.3) that was higher than other blood glucose-lowering drugs (A10B; HR = 2.1, 95% CI 2.0 to 2.2).

## Blood and blood-forming organs (ATC-group B)

Prescriptions for antithrombotic agents (B01A), containing vitamin K antagonists, platelet aggregation inhibitors, as well as direct factor Xa inhibitors, were common and associated with mortality (HR = 2.0, 95% CI 1.9 to 2.0) (Fig 1). Iron preparations (B03A) were also

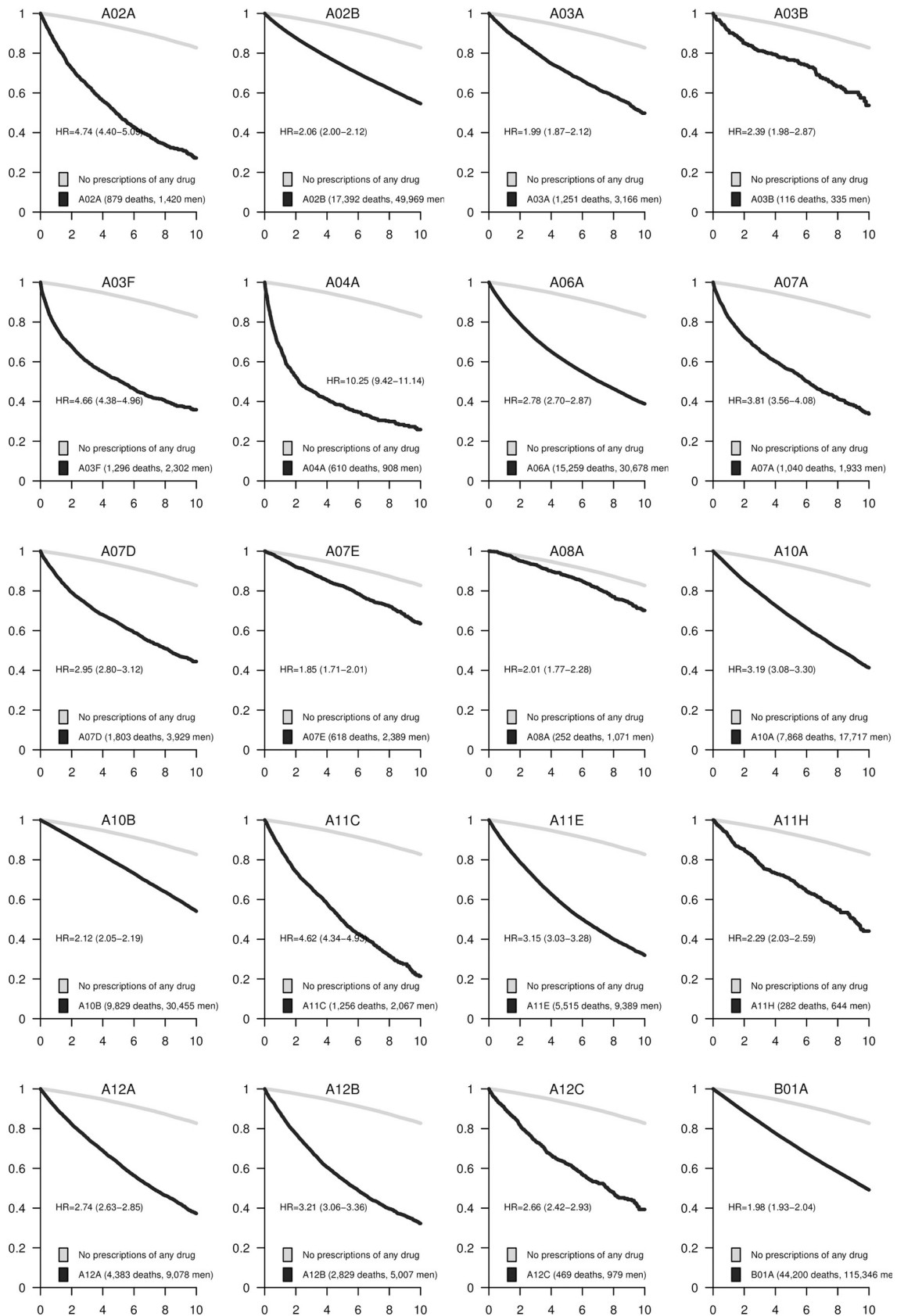

**Fig 1. Kaplan–Meier curves and estimated hazard ratios (HR) for pharmaceutical subgroups A02A-B01A.** The number of events and total number of subjects in each category are presented in the legends. 95% confidence intervals (CI) are shown for the HRs.

associated with mortality (HR = 3.5, 95% CI 3.4 to 3.7), as were vitamin B12 and folic acid (B03B; HR = 2.5, 95% CI 2.4 to 2.5; Fig 2). The strongest association (HR = 7.3, 95% CI 6.7 to 7.9) between antianemic preparations and mortality was seen in the group B03X containing, for example erythropoietin used in renal anemia in patients on hemodialysis.

## Cardiovascular system (ATC-group C)

Cardiac glucosides (C01A; Fig 2) were associated with the mortality rate (HR = 3.2, 95% CI 3.0 to 3.3), whereas class I and III antiarrhythmics (C01B), cardiac stimulants (excluding cardiac glucosides; C01C), and vasodilators (C01D; including e.g. nitrates) only displayed weak associations with mortality rate.

Prescriptions for diuretics were common and their relation to the mortality rate varied. Although thiazide diuretics (C03A) were weakly associated with the mortality rate (HR = 1.5, 95% CI 1.4 to 1.5), loop diuretics (C03C) such as furosemide had a stronger association (HR = 3.7, 95% CI 3.6 to 3.8). Moreover, potassium-sparing agents (C03D) were associated with mortality (HR = 3.6; 95% CI 3.4 to 3.7); however, combinations of diuretics and potassium-sparing agents had a weaker association (C03E; HR = 1.4, 95% CI 1.4 to 1.5).

Prescriptions for beta-adrenergic blocking agents (C07A) were prevalent and had a modest association with the mortality rate (HR = 1.9, 95% CI 1.9 to 2.0). Selective calcium channel blockers with mainly vascular effects (C08C), for example, those indicated for the treatment of hypertension, such as angiotensin-converting enzyme (ACE) inhibitors (C09A-B; Fig 3) and angiotensin II receptor blockers (C09C-D) were weakly associated with an increased mortality rate (HR ranging from 1.4 to 1.7). Drugs with direct cardiac effects (C08D) were less prevalent but were also associated with mortality (HR = 2.1, 95% CI 1.9 to 2.2). Lastly, plain lipid-modifying agents (C10A) were weakly associated with mortality (HR = 1.7; 95% CI 1.6 to 1.7).

## Dermatologicals (ATC-group D)

Antipruritics, including antihistamines and anesthetics (D04A), were associated with an increased mortality rate (HR 2.2, 95% CI 1.9 to 2.7). The association for other dermatological preparations (D11A) was weaker (HR = 1.6, 95% CI 1.4 to 1.8).

## Genito-urinary system and sex hormones (ATC-group G)

Urologicals (G04B), including acidifiers; urinary concrement solvents; drugs for urinary frequency, incontinence, and erectile dysfunction (HR = 1.4, 95% CI 1.3 to 1.4); and drugs used in benign prostatic hypertrophy (G04C; HR = 1.4, 95% CI 1.4 to 1.5) appeared to be only weakly associated with the mortality rate.

## Systemic hormonal preparations, excl. sex hormones, and insulins (ATC-group H)

Corticosteroids for systemic use (H02A) displayed an association (HR = 2.5, 95% CI 2.4 to 2.6) with mortality; however, the associations between thyroid preparations (H03A) and mortality were weaker (HR = 1.8, 95% CI 1.7 to 1.8).

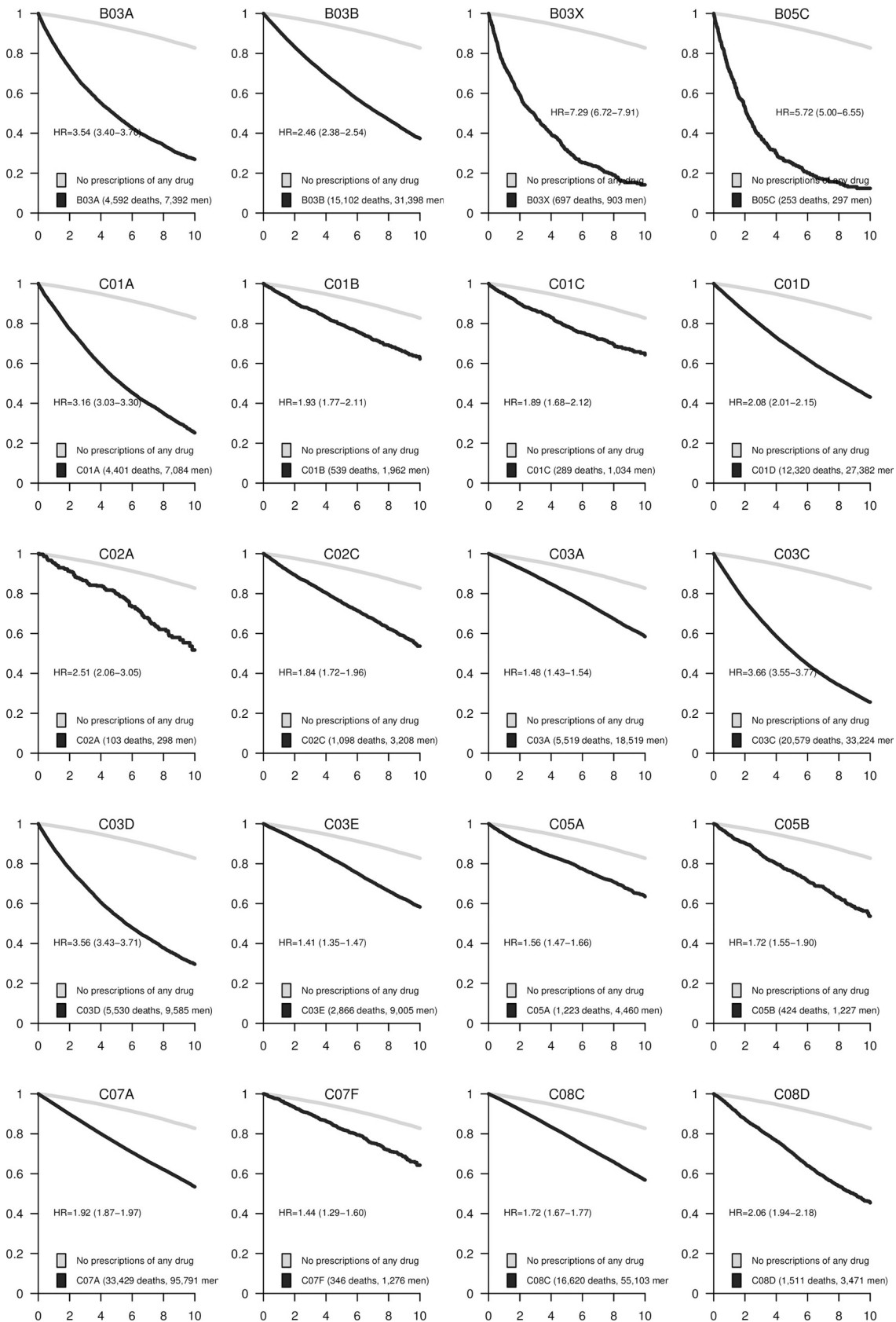

**Fig 2. Kaplan–Meier curves and estimated hazard ratios (HR) for pharmaceutical subgroups B03A-C08D.** The number of events and total number of subjects in each category are presented in the legends. 95% confidence intervals (CI) are shown for the HRs.

### Antiinfectives for systemic use (ATC-group J)

Among antibiotics, the weakest associations were seen for tetracyclines (J01A; HR = 1.9, 95% CI 1.8 to 1.9), and beta-lactam antibacterials and penicillins (J01C; HR = 1.9, 95% CI 1.8 to 2.0). Other beta-lactam antibacterials (J01D), sulfonamides and trimethoprim (J01E), macrolides, lincosamides and streptogramins (J01F), quinolones (J01M), and other antibacterials (J01X) all had a somewhat stronger association with mortality (Fig 3).

### Antineoplastic and immunomodulating agents (ATC-group L)

The strongest associations in this group were seen for alkylating agents (L01A; Fig 4; HR = 9.4, 95% CI 8.4 to 10.7), immunostimulants such as G-CSF, interferons, and interleukins (L03A; HR = 5.2, 95% CI 4.5 to 6.0), and other antineoplastic agents (L01X; HR = 4.8, 95% CI 4.3 to 5.4). Weaker associations were seen for antimetabolites (L01B; HR = 2.4, 95% CI 2.1 to 2.7) and immunosuppressants (L04A; HR = 2.5, 95% CI 2.4 to 2.7).

### Musculo-skeletal system (ATC-group M)

Antiinflammatory, antirheumatic products, and non-steroid anti-inflammatory drugs (NSAIDs and M01A) only had a weak association with mortality (HR = 1.3, 95% CI 1.3 to 1.4). Topical products for joint and muscular pain (M02A), muscle relaxants, centrally-acting agents (M03B), antigout preparations (M04A), and drugs affecting bone structure and mineralization (M05B) were all moderately associated with mortality (Fig 4).

### Nervous system (ATC-group N)

The strongest associations in this group were seen for Parkinson's disease medications, anticholinergic agents (N04A; HR = 5.2, 95% CI 4.7 to 5.8), antipsychotics (N05A; HR = 4.6, 95% CI 4.4 to 4.8), and anti-dementia drugs (N06D, Fig 5; HR = 4.9, 95% CI 4.7 to 5.2). Among analgesics, the group of analgesics and antipyretics (N02B; Fig 4) notably containing acetaminophen, had an equally strong association with mortality (HR = 2.4, 95% CI 2.3 to 2.5) as did opioids (N02A; HR = 2.3, 95% CI 2.3 to 2.4), but the strongest association in this category was seen for local anesthetics (N01B; HR = 4.1, 95% CI 3.8 to 4.4). Antiepileptics (N03A), dopaminergic agents for Parkinson's disease (N04B), anxiolytics (N05B), antidepressants (N06A), and drugs used in addictive disorders (N07B) were associated with an approximately tripled mortality rate. Somewhat weaker associations were seen for antimigraine preparations (N02C; HR = 2.0, 95% CI 1.8 to 2.2), hypnotics and sedatives (N05C; Fig 5; HR = 2.5, 95% CI 2.4 to 2.5), and psychostimulants, such as agents used for ADHD (N06B; HR = 2.3, 95% CI 1.8 to 2.9).

### Complementary analyses

Restricting the analyses to men with CCI = 0 generated similar results; however, the estimated associations tended to be overall weaker (S1–S5 Figs). Adjusting for age as a continuous and quadratic term in the Cox models instead of stratifying for age produced nearly identical HRs (data not shown).

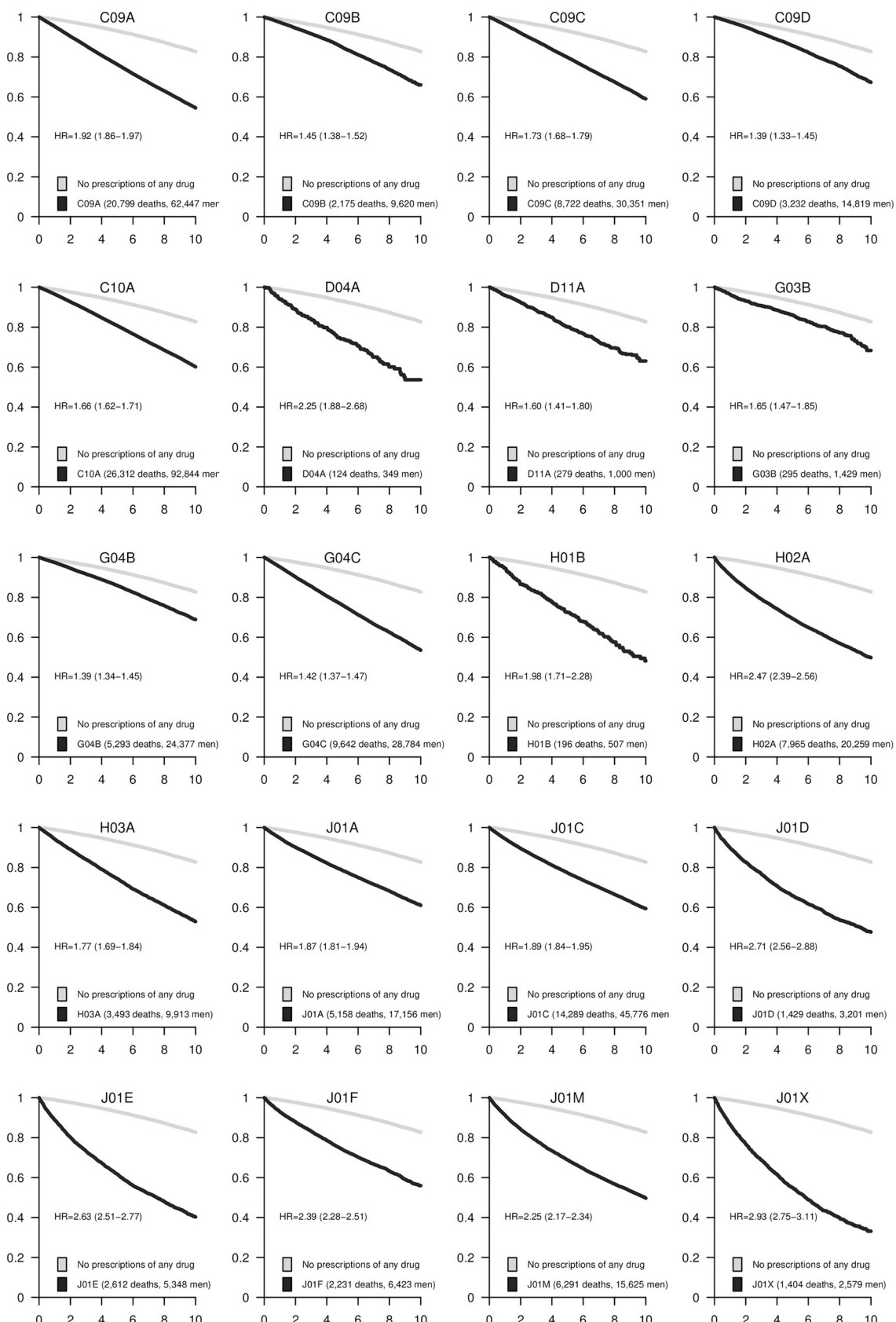

**Fig 3. Kaplan–Meier curves and estimated hazard ratios (HR) for pharmaceutical subgroups C09A-J01X.** The number of events and total number of subjects in each category are presented in the legends. 95% confidence intervals (CI) are shown for the HRs.

## Discussion

In this large cohort of elderly men randomly selected from the general population as controls to men with prostate cancer, 73% were categorized as free of comorbidity, according to the CCI. Nevertheless, 84% of the men had at least one prescription filled during the year before the start of follow-up. Many ATC-codes are found to be associated with mortality, as is expected from the severity of the conditions for which they are indicated. However, drug classes that were not directly linked to a specific comorbidity were also found to be associated with an increased risk of death. In the present study, contrary to expectations, some drugs previously associated with mortality were only weakly associated with an increased risk of death. In contrast, other drugs had unexpectedly strong associations with an increased risk of death in the present study. These findings suggest that analysis of ATC-codes might provide relevant information regarding the patient's health status even if the indication is unknown or not associated with mortality risk per se.

In the present study, the majority of participants had filled at least one prescription in the year preceding follow-up, even though the CCI classified them as comorbidity-free. This finding is in line with a previous study showing that in two different datasets, 68% and 74% of hospitalized patients, respectively, had at least one prescription without a corresponding ICD-10 code for a disease entity related to the use of the drug [6]. This phenomenon provides an opportunity to further characterize baseline risk in a large proportion of individuals considered free of comorbidity, according to the widely used CCI.

Prescription data is widely available in health data registries or from electronic health records. It is a well-structured type of information that can be incorporated into primary data collection. A further advantage is that this type of data source covers prescriptions from both tertiary and primary care.

Prescription data has been used to capture comorbidity in previously-reported models [5, 14–16]. The strategy for using this type of information has most often involved defining the comorbid conditions of interest and mapping medications related to these comorbidities [6, 17]. We propose to instead explore information provided by every issued prescription, without assessing their potential value or relation to any specified comorbidity. From this perspective, the results from this study are of interest.

In principle, there might be different reasons behind an association between a class of prescribed drugs and mortality. The first consideration is if the medication in itself is causally related to the estimated mortality risk. However, more likely, the prescription indicates a patient's health condition, which impacts mortality risk. In other words, it is the indication and not the drug itself that affects the risk of death.

Nevertheless, in several instances, the results of this study demonstrated unexpectedly strong associations with respect to health status assessment. For example, a 2–5 times increase in the mortality rate associated with the use of antacids is unlikely to reflect the mortality directly related to gastrointestinal bleeding. The association was strongest for locally acting antacids, and their use could reflect the patients' overall comorbidity burden and frailty. The antacids group also contains bicarbonate used in advanced chronic renal failure, a condition associated with high mortality.

In this context, granularity of data might be paramount. The present study's findings pertaining to use of diuretics illustrate this idea well. In our dataset, loop diuretics and potassium-

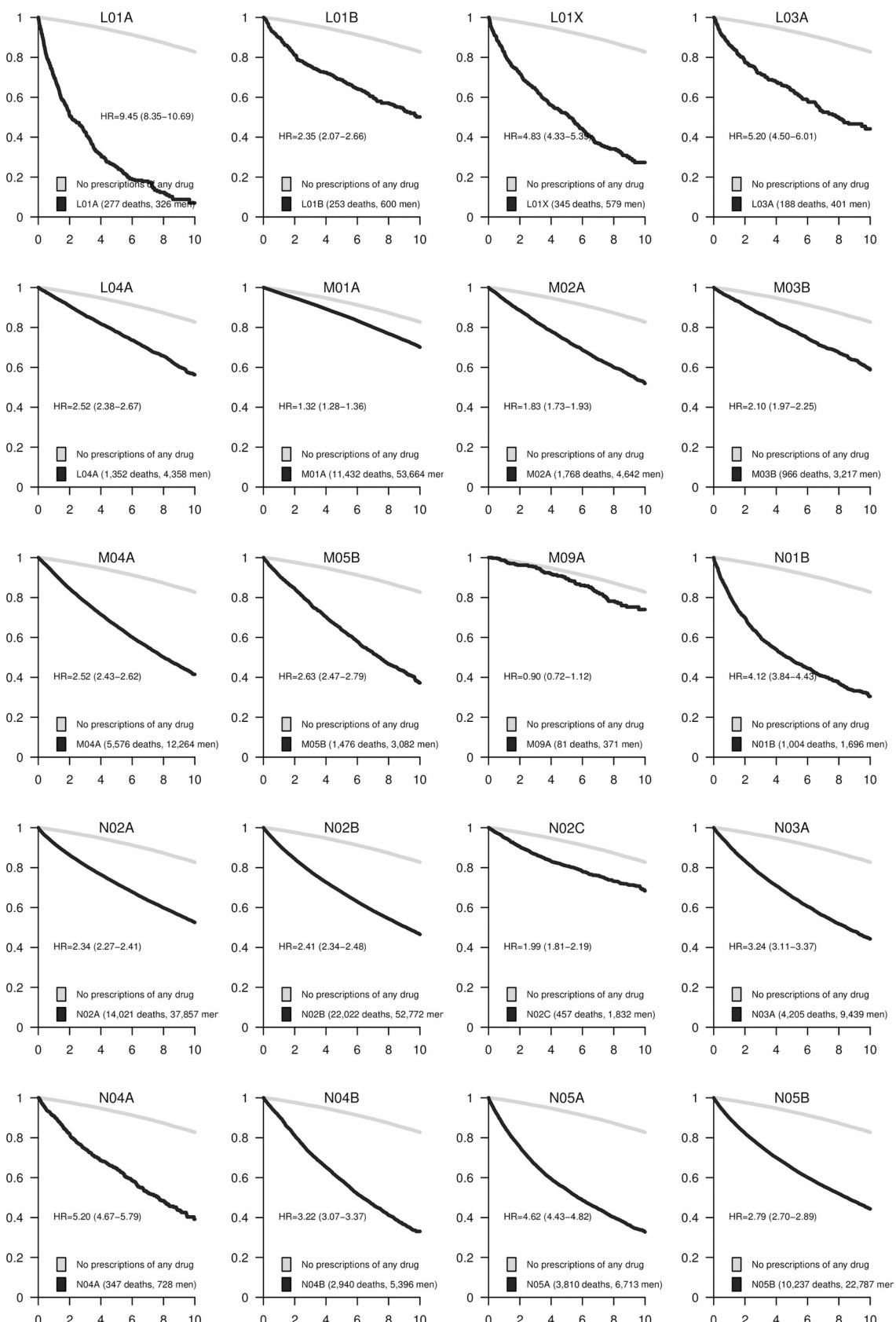

**Fig 4. Kaplan–Meier curves and estimated hazard ratios (HR) for pharmaceutical subgroups L01A-N05B.** The number of events and total number of subjects in each category are presented in the legends. 95% confidence intervals (CI) are shown for the HRs.

sparing agents were strongly associated with the mortality rate, whereas thiazide diuretics and diuretics combined with potassium-sparing agents had weaker associations. Therefore, considerable variations occur within the drug class of diuretics. This is expected since loop diuretics are prescribed in association with more severe conditions, such as heart and kidney failure, than thiazide diuretics are. This underlines the importance of information carried in the third level ATC code, specifically, the pharmacological subgroup, over the information carried by the first level code, i.e. the main anatomical group.

In the present study, symptomatic treatments without a strictly defined disease indication were strongly associated with mortality rate, likely reflecting poor general health. Nausea and vomiting are not life-threatening per se, but when prescription medication is needed to alleviate such symptoms, it might signal associated conditions, such as malignancy and associated treatment, or general frailty. In the present study, such estimated associations were particularly strong for serotonin antagonists. The analgesic, acetaminophen, had a strong association with the mortality rate and was comparable to that of opioids. Hypothetically, the type of medication combined with the duration of use or repeated use may provide further information about an individual's frailty.

Other unexpected associations were observed. For example, the association between use of vitamin A and D and mortality was stronger compared to the association between insulin and mortality. This association likely reflects general frailty. Similarly, supplementation with potassium that is mainly prescribed in association with chronic use of diuretics, such as in cardiac failure and calcium deficiencies, is perhaps mainly done in frail older adults with osteoporosis.

Prescription of a drug can also indicate that the patient has a relatively low mortality risk. For example, in the present study, NSAIDs were only weakly associated with an increased mortality rate. However, as risk minimization measures have been introduced in relation to the use of NSAIDs in patients with cardiovascular disease [18–20], this weak association likely reflects the selection of a subgroup of patients that are sufficiently healthy or strong to receive such treatment.

For some pharmaceutical subgroups the association with mortality was unexpectedly weak. Most major cardiovascular conditions were expected to be associated with mortality. While some drugs in this category are commonly prescribed for hypertension and, therefore, not expected to be strongly associated with mortality; unexpected weak associations were seen for class I and III antiarrhythmics, cardiac stimulants (excluding cardiac glucosides; C01C), and vasodilators (including nitrates). Speculatively, prescribers are aware of risks associated with the use of antiarrhythmics, and therefore, use of these drugs is largely avoided in frail patients.

An important limitation of the prescription registry is that it does not cover medicines administered in-hospital. For example, cardiac stimulants (C01C) are used in severe cardiac failure and are likely to be associated with mortality rate. However, these drugs were not associated with mortality rate in this study, likely due to the fact that these drugs are only administered intravenously to hospitalized patients, and hence not recorded in the Swedish prescribed drug registry.

The results from this study might not be directly generalizable to other settings since prescription patterns vary between countries and regions, and over time. The information carried by prescriptions might also to some extent vary depending on the population studied. In the present study, the population of elderly men free of prostate cancer represented a specific subset of the general population. For example, there was an association between use of iron and

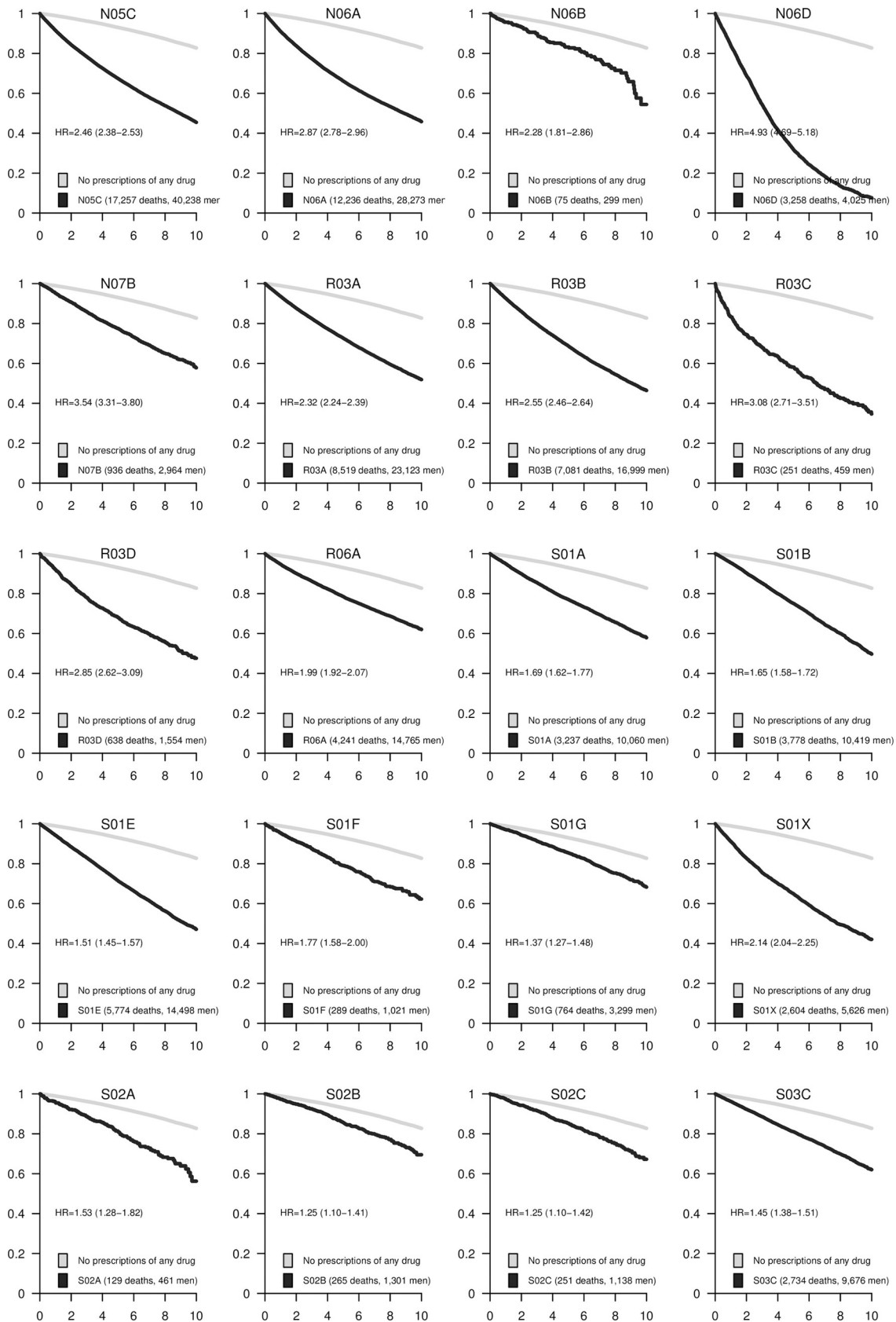

**Fig 5. Kaplan–Meier curves and estimated hazard ratios (HR) for pharmaceutical subgroups N05C-S03C.** The number of events and total number of subjects in each category are presented in the legends. 95% confidence intervals (CI) are shown for the HRs.

mortality rate in our study, suggesting existence of indications for iron supplementation that are otherwise associated with increased risk of mortality in men. However, in a population of younger women, there could be indications for iron supplementation that are not associated with increased risk of mortality, such as menstruation. Another example is the use of antiandrogens (G03H) that could be expected to be associated with mortality, but not in a population selected as free of prostate cancer at baseline. Since our research focuses on the epidemiology of prostate cancer, our study population was restricted to males and essentially excluded men without prostate cancer, which is a notable limitation for generalization of study results. These observations of potentially important influence of population-selection suggest that a population- and context-specific model might be warranted when using this type of information to estimate baseline risk in practice. This should be further explored in future research within this field.

In conclusion, in the present study, the majority of elderly men had a history of being prescribed medications associated with an increased mortality rate. This was also seen for drug classes not indicated for a specific comorbidity associated with increased risk of death. It may therefore be warranted to explore the value of patients' overall prescription history as a measure of comorbidity burden, rather than restricting such analyses to medications related to specific comorbidities. Such approach might need to be adapted to the specific context where it is used.

## Supporting information

**S1 Table. All ATC-codes at the level of pharmaceutical subgroups (having four positions in the code) observed in the study database.** The number of men having at least one mentioning of the code during the year preceding the index date, and the number of those that died, are provided. In PCBaSe prescriptions from all anatomical main groups A, B, C, D, G, H, J, L, M, N, P, R, S and V are available, except the pharmacological subgroups A01, A05, A09, A16 B02, B06 D01, D02, D03, D05, D06, D07, D08, D09, D10 G01, G02 J02, J04, J05, J06, J07 P01, P02, P03 R01, R05, V01, V03, V04, V07, and V08.
(DOCX)

**S1 Fig. Kaplan–Meier curves and estimated hazard ratios (HR) for pharmaceutical subgroups A02A-B01A.** The number of events and total number of subjects in each category are presented in the legends. 95% confidence intervals (CI) are shown for the HRs. The analysis has been restricted to men with CCI = 0.
(PDF)

**S2 Fig. Kaplan–Meier curves and estimated hazard ratios (HR) for pharmaceutical subgroups B03A-C08D.** The number of events and total number of subjects in each category are presented in the legends. 95% confidence intervals (CI) are shown for the HRs. The analysis has been restricted to men with CCI = 0.
(PDF)

**S3 Fig. Kaplan–Meier curves and estimated hazard ratios (HR) for pharmaceutical subgroups C09A-J01X.** The number of events and total number of subjects in each category are presented in the legends. 95% confidence intervals (CI) are shown for the HRs. The analysis

has been restricted to men with CCI = 0.
(PDF)

**S4 Fig. Kaplan–Meier curves and estimated hazard ratios (HR) for pharmaceutical sub-groups L01A-N05B.** The number of events and total number of subjects in each category are presented in the legends. 95% confidence intervals (CI) are shown for the HRs. The analysis has been restricted to men with CCI = 0.
(PDF)

**S5 Fig. Kaplan–Meier curves and estimated hazard ratios (HR) for pharmaceutical sub-groups N05C-S03C.** The number of events and total number of subjects in each category are presented in the legends. 95% confidence intervals (CI) are shown for the HRs. The analysis has been restricted to men with CCI = 0.
(PDF)

## Acknowledgments

We would like to thank Editage (www.editage.com) for English language editing.

**Disclaimer:** Rolf Gedeborg is also employed by the Medical Products Agency (MPA) in Sweden. The MPA is a Swedish Government Agency. The views expressed in this article may not represent the views of the MPA.

## Author Contributions

**Conceptualization:** Rolf Gedeborg, Hans Garmo, David Robinson, Pär Stattin.

**Data curation:** Hans Garmo.

**Formal analysis:** Hans Garmo.

**Funding acquisition:** Pär Stattin.

**Investigation:** Hans Garmo.

**Methodology:** Rolf Gedeborg, Hans Garmo, Pär Stattin.

**Project administration:** Rolf Gedeborg, Pär Stattin.

**Resources:** David Robinson, Pär Stattin.

**Supervision:** David Robinson, Pär Stattin.

**Validation:** Hans Garmo.

**Visualization:** Hans Garmo.

**Writing – original draft:** Rolf Gedeborg, Hans Garmo, David Robinson, Pär Stattin.

**Writing – review & editing:** Rolf Gedeborg, Hans Garmo, David Robinson, Pär Stattin.

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
