## [Decision Letter · Decision Letter 0]

21 Aug 2020

PONE-D-20-18083

Prescription-based prediction of baseline mortality risk among older men

PLOS ONE

Dear Dr. Gedeborg,

Thank you for submitting your manuscript to PLOS ONE. After careful consideration, we feel that it has merit but does not fully meet PLOS ONE’s publication criteria as it currently stands. Therefore, we invite you to submit a revised version of the manuscript that addresses the points raised during the review process.

We apologize for the delay in the review process. It has unfortunately been hard to find reviewers in COVID-times. After obtaining a real good one (as per below) I therefore reviewed it thoroughly myself for the second review.

We find you article of interest but it would require major changes before it would be judged suitable for publication. 1)      I disagree that 16% of the population and indeed 52,677 individuals are too few to be the main comparison group. I would much prefer them to be used as the reference  - it would certainly add consistency and general usefulness to the analyses. Please change the reference group.2)      The methods section does not describe in detail in what time window during the study the prescriptions were made. Using “anytime during follow-up” would load massive amounts of immortal-time bias - especially in that you now use “individuals with at least one prescription” as the reference group. Please state how you defined prescription in terms of time and make sure all prescriptions preceded study entry. (Preferably all prescriptions filed the year (or two years) before study entry – this would also improve the usefulness of the methods in other epidemiological studies). Or alternativly in a time-variant manner (even though I believe most researchers would not use that in research practice for adjustment purposes).3)      The discussion lacks a discussion of why you did not perform the study in a population including women and potential lack of generalizability due to this fact.4)      Please add analyses of “any drug prescription” as suggested by reviewer 1.5)      The discussion of furosemide. You need to add that the association is likely due to the drug being commonly prescribed in “palliative cancer, heart failure etc” patients to relieve swollen legs and pulmonary edema. It is therefore not at all surprising that it is associated to poor survival.6)      I think your findings on cardiovascular drugs is because indeed individuals with hypertension only tend to end-up in this ATC-group and they may be very healthy and health conscious. It may also be an artefact of using the “some other drug” reference group. Please discuss, there are several references on claims data and hypertension on this topic.Consider using age as a continuous and quadratic in your analyses - it is the strongest known predictor of mortality.In table 1 CCI=0 the number (n=23,7515) should be changed to 237,515.

We look forward to receiving your revised manuscript.

Kind regards,

Louise Emilsson

Academic Editor

PLOS ONE

Journal Requirements:

Reviewers' comments:

Reviewer's Responses to Questions

**Comments to the Author**

1. Is the manuscript technically sound, and do the data support the conclusions?

Reviewer #1: Yes

2. Has the statistical analysis been performed appropriately and rigorously? 

Reviewer #1: Yes

3. Have the authors made all data underlying the findings in their manuscript fully available?

Reviewer #1: Yes

4. Is the manuscript presented in an intelligible fashion and written in standard English?

Reviewer #1: Yes

5. Review Comments to the Author

Reviewer #1: Gedeborg et al. Prescription-based prediction of baseline mortality risk among older men

Abstract.

1. The authors want to offer a method that is better than Charlson to predict mortality.

If so it would be nice to see:

a) prescription data limited to patients with Charlson = 0. What does prescription data add there?

b) and a model where both Charlson index + Prescriptions are added, how much does the prescription data add then?

2. could the men in question have other types of cancers than prostate cancers?

******

introduction.

well-written. Relevant papers are cited.

While this may be the first study to explore mortality according to ”all prescribed drugs” it cannot possible be the first to describe ”all prescription” and comorbidity. I know that manu authors use ” number of drugs prescribed last 12 months” as a covariate for their comorbidity research.

Gedeborg et al should acknowledge that and mention that in their paper.

methods

did the research ethics review really approve the study protocol? you had a detailed study protocol?

Please describe the Prescribed Drug Register with a few words (+ when it began), and what drugs are not registered? and that inpatient drugs are not covered.

you counted the number of prescriptions. is that the most natural way to look at comorbidity? does ot dosage play a role.

Results

It would be natural to present the overall HR for any prescription and risk of death.

1 drug

2 drugs

3 drugs

4 drugs or more…

Discussion. the association between several of the drugs and mortality, is ot strange considering what these drugs are prescribed for, add a few sentences on that-

Please note both erythropoietin and atipropulsives are sometimes used as part of cancer management (that is when I use it myself.

The long discussion (results) of different pharma categoreies is fine.

I miss a discussion about. residual confoundig, add text, and also comment more on the limitations of this paper.

also add:

I would like to see some more emphasis on "number of drugs" and comorbidity rather than just different types of dugs (ATC groups).

HRs for death according to 0 drug [ref], 1 drug, 2 drugs, 3 drugs etc last year.

Cause-specific death according to number of drugs.

How much does the addition of one drug shorten the life expectancy, in a say 70-yr-old person? HRs for death according to 0 drug [ref], 1 drug, 2 drugs, 3 drugs etc last year.

Cause-specific death according to number of drugs.

How much does the addition of one drug shorten the life expectancy, in a say 70-yr-old person?

6. PLOS authors have the option to publish the peer review history of their article (what does this mean?). If published, this will include your full peer review and any attached files.

Reviewer #1: No

---

## [Author Response · Author response to Decision Letter 0]

25 Sep 2020

Please see separate response letter attached.

---

## [Decision Letter · Decision Letter 1]

15 Oct 2020

Prescription-based prediction of baseline mortality risk among older men

PONE-D-20-18083R1

Dear Dr. Gedeborg,

We’re pleased to inform you that your manuscript has been judged scientifically suitable for publication and will be formally accepted for publication once it meets all outstanding technical requirements.

Kind regards,

Louise Emilsson

Academic Editor

PLOS ONE

Additional Editor Comments (optional):

Great revision and responses to our previous comments. Looking forward to see this in print. 

Reviewers' comments:

Reviewer's Responses to Questions

**Comments to the Author**

1. If the authors have adequately addressed your comments raised in a previous round of review and you feel that this manuscript is now acceptable for publication, you may indicate that here to bypass the “Comments to the Author” section, enter your conflict of interest statement in the “Confidential to Editor” section, and submit your "Accept" recommendation.

Reviewer #1: All comments have been addressed

2. Is the manuscript technically sound, and do the data support the conclusions?

Reviewer #1: Yes

3. Has the statistical analysis been performed appropriately and rigorously? 

Reviewer #1: Yes

4. Have the authors made all data underlying the findings in their manuscript fully available?

Reviewer #1: Yes

5. Is the manuscript presented in an intelligible fashion and written in standard English?

Reviewer #1: Yes

6. Review Comments to the Author

Reviewer #1: No comment

No comment

No comment

No comment

No comment

No comment

No comment

No comment

No comment

No comment

No comment

No comment

7. PLOS authors have the option to publish the peer review history of their article (what does this mean?). If published, this will include your full peer review and any attached files.

Reviewer #1: No

---

## [Editor Report · Acceptance letter]

19 Oct 2020

PONE-D-20-18083R1 

Prescription-based prediction of baseline mortality risk among older men 

Dear Dr. Gedeborg:

I'm pleased to inform you that your manuscript has been deemed suitable for publication in PLOS ONE. Congratulations! Your manuscript is now with our production department. 

Kind regards, 

on behalf of

Dr. Louise Emilsson 

Academic Editor

PLOS ONE